# Building clone-consistent ecosystem models

**Gerrit Ansmann**[1]*, **Tobias Bollenbach**[1,2]

**1** Institute for Biological Physics, University of Cologne, Cologne, Germany, **2** Center for Data and Simulation Science, University of Cologne, Cologne, Germany

* gansmann@uni-koeln.de

## Abstract

Many ecological studies employ general models that can feature an arbitrary number of populations. A critical requirement imposed on such models is *clone consistency*: If the individuals from two populations are indistinguishable, joining these populations into one shall not affect the outcome of the model. Otherwise a model produces different outcomes for the same scenario. Using functional analysis, we comprehensively characterize all clone-consistent models: We prove that they are necessarily composed from basic building blocks, namely linear combinations of parameters and abundances. These strong constraints enable a straightforward validation of model consistency. Although clone consistency can always be achieved with sufficient assumptions, we argue that it is important to explicitly name and consider the assumptions made: They may not be justified or limit the applicability of models and the generality of the results obtained with them. Moreover, our insights facilitate building new clone-consistent models, which we illustrate for a data-driven model of microbial communities. Finally, our insights point to new relevant forms of general models for theoretical ecology. Our framework thus provides a systematic way of comprehending ecological models, which can guide a wide range of studies.

**Data Availability Statement:** All relevant data is within the manuscript and its Supporting information.

**Funding:** This work was supported in part by German Research Foundation (DFG) Collaborative Research Centre (SFB) 1310 (to T.B.), and Austrian

## Author summary

Mathematical models of population dynamics are an important tool to advance our understanding of ecosystems, which can be relevant for environmental, clinical, and industrial applications. One sanity check for such models is to virtually split a population into two with identical properties – allegorically, we paint half the individuals of the population in a different color. As we do not change the ecological situation, the outcome of the model should not change either; we call this feature *clone consistency*. We investigated the mathematical properties of clone-consistent models and deduced simple rules for their form. These rules allow to easily check clone consistency in existing models and ensure it when building new ones. The resulting framework can guide researchers in building models for specific ecosystems and in investigating general properties of ecosystems. We showcase our approach by applying it to models for bacterial communities causing urinary-tract infections. We further discuss that clone inconsistency, which occurs in several prominent models, reflects strong, often implicit, assumptions and it is important to check whether these are justified. Such assumptions

Science Fund (FWF) standalone grant P 27201-B22 (to T.B.). The funders had no role in study design, data collection and analysis, decision to publish, or preparation of the manuscript.

**Competing interests:** The authors have declared that no competing interests exist.

may diminish the applicability of these models and the generality of results obtained with them.

This is a *PLOS Computational Biology* Methods paper.

## Introduction

Many theoretical and semi-empirical studies of ecological communities employ general models that are not specific to a given community, but can incorporate an arbitrary number of populations with different properties [1–4]. In most such models, the equations governing each population have the same form, and the species of a population only manifests in the values of the associated parameters. These parameters may describe the properties of a single population, the interaction of two populations, or higher-order interactions, i.e. effects involving three or more populations [5, 6]. Interaction parameters are often chosen randomly [7–13] or determined from experiment [14–17].

Developing such models is one of the challenges of modern ecology, in particular when incorporating empirical data [18]. For example, recent advances in automating experiments have enabled measuring interaction parameters for richer communities [16, 19–21], interactions characterized by more than one observable [16], or higher-order interactions [15, 22]. These new experimental scenarios call for new ecological models that can incorporate the respective data. Existing models are often not suitable here since there is no uniform answer as to how multi-parameter or higher-order interactions should be measured [3, 6, 16, 20, 23]. Another driver of new modeling approaches is growing computational power [18, 24], which allows to investigate increasingly general and complex models [12, 25].

To improve the modeling process, several collections of criteria capturing consistency were suggested [26–34]. While many of these are specific to the ecological scenario considered, e.g., predation, the following invariance is a recurring theme [28–38]: If two populations have identical parameter values, they contain identical individuals (clones) within a general model. Thus, the outcome of the model must only depend on the total abundance of these two populations, and not on how the clones are assigned to them. We call this criterion *clone consistency*. Similar criteria for models have been named *invariance under relabeling* [32] or *under identification/aggregation of identical species* [29–31] as well as *"common-sense" condition* [34, 36, 38]. Further, it is often required that joining two populations of identical individuals does not affect diversity measures and other ecological observables [39–41]; this concept was introduced under the name *twin property* [42]. In the analysis and modeling of food webs, this issue is essentially circumvented by considering *trophic species*, which aggregate species with identical predators or prey – a controversial approach [43–47]. Finally, if a model is clone-consistent and clones actually exist, it can be simplified; this is called *aggregating* or *lumping* [48, 49].

To provide an instructive example for clone inconsistency, we compare two simulations of a predator–prey scenario using the same general model [51] (chosen here exclusively for its simplicity):

$$\dot{x}_j = x_j\left(a_j + \sum_i b_{ji} \log\left(x_i\right)\right),\tag{1}$$

where $x_j$ is the abundance of population $j$, and $a$ and $b$ are parameters governing the properties

and interaction of the populations. In the first simulation (solid lines in Fig 1), we chose:

$$a = \begin{pmatrix} 0.3 \\ -0.3 \end{pmatrix} \quad \text{and} \quad b = \begin{pmatrix} 0.5 & -1.0 \\ 0.8 & -0.5 \end{pmatrix}. \tag{2}$$

Population 1 is the prey; Population 2 contains the predators. In the second simulation (dashed lines in Fig 1), we split the predator population into two sub-populations (2 and 3) with identical properties and half the initial abundance. Allegorically, we paint half the predators in a different color. In numbers:

$$a = \begin{pmatrix} 0.3 \\ -0.3 \\ -0.3 \end{pmatrix} \quad \text{and} \quad b = \begin{pmatrix} 0.5 & -1.0 & -1.0 \\ 0.8 & -0.5 & -0.5 \\ 0.8 & -0.5 & -0.5 \end{pmatrix}. \tag{3}$$

Although these two simulations describe the same situation, their outcomes differ strongly (Fig 1): Not only does the amplitude of the predator and prey abundances change, but the type of population dynamics changes from an oscillation to a simple convergence on a fixed point. These incompatible results show that the model used in these simulations suffers from a fundamental inconsistency. In this work, we show that this is a consequence of the logarithm being used in this way and not being additive, i.e., that $\log(y) + \log(z) \neq \log(y + z)$.

While splitting populations is an illustrative thought experiment, its implications reach farther for at least three reasons: First, in actual modeling we can encounter the inverse situation, i.e., two populations with identical properties. Second, if there are problems when two populations have absolutely identical properties, there will also be problems when they have similar properties since models, like nature, are continuous. Third, problems can already arise if two populations are similar in one aspect that is relevant to the model. In a variation of the above

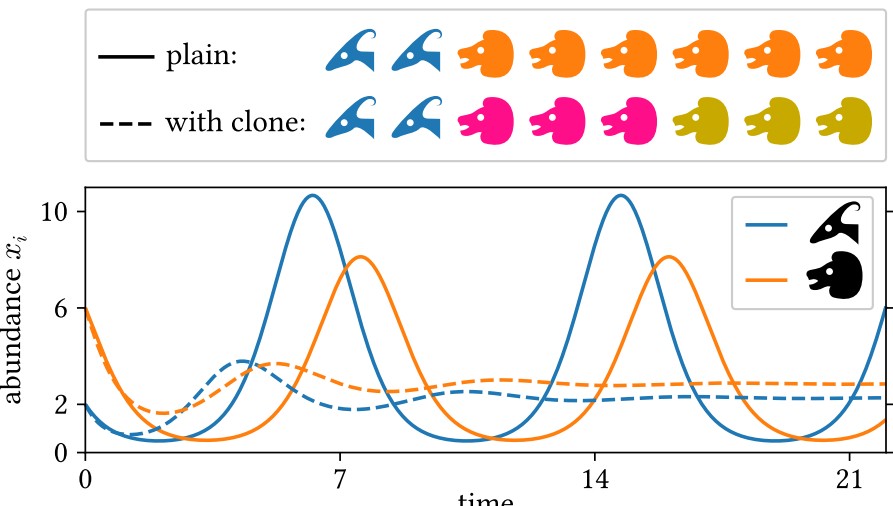

**Fig 1. Example of clone inconsistency: Two simulations of a simple predator–prey scenario using the same general model (see text for details).** Solid lines: simulation using one population for prey (blue, antelopes) and predators (orange, lions) each. The initial abundances are $x_1(0) = 2$ and $x_2(0) = 6$ (animal heads in the top legend). Dashed lines: same, but with two identical predator sub-populations (pink and ocher) with half the initial abundance; the abundance shown for the predators is the sum over the two sub-populations. The simulations were run with JiTCODE [51] using the DoPri5 method (S1 Code).

example, if the members of two populations of predators prey with a similar rate on a given focal species, assigning individual predators to the other population should not disproportionately affect the total predation rate – even when these predators reproduce at different rates.

Despite its simplicity, ensuring clone consistency directly can be tedious as it requires finding a counter-example or performing a model-specific proof. For example, Morozov and Petrovskii [33] spent several pages of calculations on checking a weaker criterion for a handful of models. Several proposed models [3, 16, 50, 52–63] are not clone-consistent (see *Implications* for details).

Here, we present a framework for checking and ensuring clone consistency in models. This paper is structured as follows: In *What models are clone-consistent?*, we introduce basic concepts and criteria on which we then build our framework employing methods from functional analysis. We explain how to use this framework to systematically assess and build models. In *Case Study*, we demonstrate the features of our framework by applying it to a recent model for microbial communities. In *Implications*, we discuss the consequences of our results. In particular, we explain how specific assumptions may fix an apparent clone-inconsistency. Our framework guides modellers by forcing them to make such assumptions explicit, which automatically raises the question if they are justified. Moreover, we discuss how our framework applies to all models involving multiple populations and its general implications for ecological modeling studies. In *Methods*, we provide elaborations and proofs for the mathematically inclined reader.

## Results/Discussion

### What models are clone-consistent?

In this section, we first define *impact functions*, which are fundamental ingredients of ecosystem models that allow us to mathematically encode our consistency criteria. We then expose the consequences of our consistency criteria – first for impact functions and then for entire models. In particular, we present recipes for checking and building models.

**Defining and constraining impact functions.** *Impact functions* describe the impact of a community on a species, on a resource, or on any other relevant feature of the ecosystem that is captured in a model. Features and phenomena described by impact functions include:

- the effective growth rate of a given species,

- the remaining size of a niche,

- the rate of predation,

- the availability of a resource or, if the resource is a dynamical variable, its consumption and production,

- reproductive services, e.g. pollination,

- the amount of crowding, and

- general interaction terms, e.g. the sum in the generalized Lotka–Volterra model [1].

The arguments of impact functions are the abundances of all populations in the ecosystem $\mathbf{x} = (x_1, x_2, \ldots, x_n)$ and parameters $\mathbf{a} = (a_1, a_2, \ldots, a_n)$ that quantify the impact of the populations. Often the impact of a population $i$ is described by a single number. Yet our results also hold for the more general case that $m$ parameters per population are required, i.e., $a_i = (a_{i1}, a_{i2}, \ldots, a_{im}) \in \mathbb{R}^m$.

A prominent example of an impact function is:

$$\mathbf{x}, \mathbf{a} \mapsto \sum_{i=1}^{n} a_i x_i, \tag{4}$$

where $\mathcal{Y} \mapsto \mathcal{Z}$ denotes the function that maps $\mathcal{Y}$ to $\mathcal{Z}$. This is employed for the interaction term in the generalized Lotka–Volterra model [1] among others. In general, impact functions can take many forms.

We require impact functions to fulfill the following basic criteria (illustrated in Fig 2; see Methods, *The functional algebra of impact functions* for mathematical formulations):

**I1** Commutativity: The properties and idiosyncrasies of a given population are exclusively captured by its associated parameters, as opposed to dedicated mathematical terms in the function. This is equivalent to pairs of abundances and parameters (($x_i, a_i$)) being interchangeable as arguments of the impact function.

**I2** When a population is absent, its associated parameters have no effect on the value of the impact function.

**I3** When each parameter associated with a given population is zero, that population's abundance has no effect on the value of the impact function.

**I4** Clone consistency: If two (or more) populations have identical parameters, the value of the impact function must only depend on their summed abundance and not on its distribution among the two populations.
Note that this criterion is a special case of clone consistency as described in the introduction. Using impact functions complying with this criterion is therefore necessary for a model to be clone-consistent, but not sufficient (we will address clone consistency on the scope of the entire model later). Due to the narrower scope of this criterion, the populations do not need to be identical in all respects, but only in the parameter(s) used by the respective impact function.

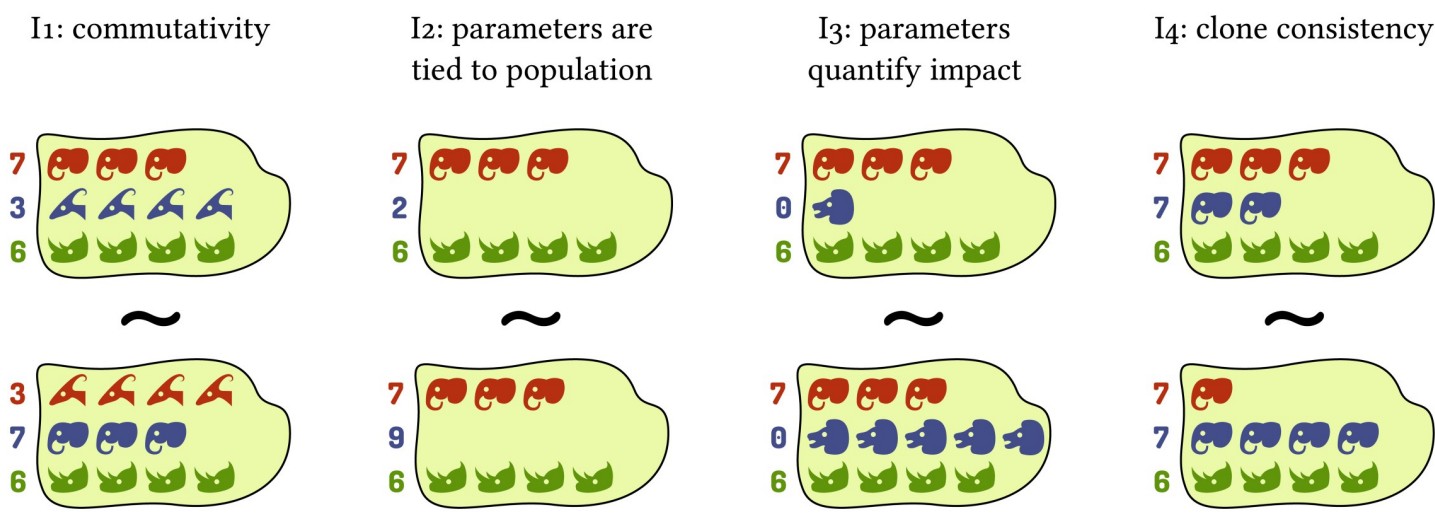

**Fig 2. Criteria for impact functions exemplified for the grazing impact of animal populations.** Each island represents a community. Each row and color represents one population in the model, with animal heads representing individuals. Numbers on the left represent parameters governing the respective population (grazing rate in the example), and head shapes indicate whether populations have identical properties as per these parameters. The similar sign (∼) indicates that two communities are equivalent as arguments of an impact function, i.e., they should yield the same result (total amount of grazing in the example).

Note that it is often reasonable to choose a considerable portion of parameters to be zero. For example, if our impact function describes predation loss of a given focal species, we would choose $a_i = 0$ for all populations $i$ that do not prey upon the focal species. In the following we call a function *impact function* if and only if it satisfies these criteria.

**The form of impact functions.** While the criteria I1–I4 for impact functions are conceptually simple, it is not straightforward to test directly if a given function complies with them or to devise a new function that does. To address this issue, we investigated the functional form of impact functions and found a set $\Omega$ of *basic impact functions*, which can serve as building blocks for ecosystem models.

These basic impact functions are linear combinations of abundances and (potentially transformed) parameters, i.e., all functions of the form:

$$\mathbf{x}, \mathbf{a} \mapsto \sum_{i=1}^{n} \zeta(a_i) x_i, \tag{5}$$

where $\zeta : \mathbb{R} \to \mathbb{R}$ is an arbitrary parameter transformation with $\zeta(0) = 0$. Often, $\zeta$ is the identity function (id), simplifying Eq 5 to Eq 4. Using methods from functional analysis, we mathematically proved that any impact function can be built from these basic impact functions via addition, multiplication, function composition, and similar operations (see Methods, *The functional algebra of impact functions*). Conversely, everything built from those elements or other impact functions is again an impact function. Formally, the general functional form of impact functions is:

$$\mathbf{x}, \mathbf{a} \mapsto \chi(\omega_1(\mathbf{x}, \mathbf{a}), \dots, \omega_l(\mathbf{x}, \mathbf{a})), \tag{6}$$

where $\omega_1, \dots, \omega_l \in \Omega$ are basic impact functions as per Eq 5 and $\chi : \mathbb{R}^l \to \mathbb{R}$ is an arbitrary function combining their results, such as a product, sum, or a more complex function.

To illustrate the composition of impact functions from these building blocks, we consider the case of a single population of flowering plants that may be both pollinated and grazed upon by several insect populations [64] as a toy example (also see Fig 3, bottom). We use a function $\rho_{\text{graz}}$ to describe the rate at which insects (and their larvae) graze on the plants:

$$\rho_{\text{graz}}(\mathbf{x}, \mathbf{a}) := \sum_{i=1}^{n} a_i x_i, \tag{7}$$

where $a_i$ is the grazing rate of insect population $i$. The function $\rho_{\text{graz}}$ has the form of Eq 5 (with $\zeta = \text{id}$) and therefore it is a (basic) impact function. We use a function $\rho_{\text{poll}}$ to capture the rate at which insects fertilize flowers:

$$\rho_{\text{poll}}(\mathbf{x}, \mathbf{b}) = \frac{1}{1 + \left(\sum_{i=1}^{n} b_i x_i\right)^{-1}}, \tag{8}$$

where $b_i$ is the contribution to pollination provided by insect population $i$. A Holling type-II response here ensures that the fertilization rate saturates at 1. Again, we can see that this is an impact function as it is built via composing $z \mapsto \frac{1}{1+z^{-1}}$ and the basic impact function $\tau(\mathbf{x}, \mathbf{b}) := \sum_{i=1}^{n} b_i x_i$. Finally, we combine the two impact functions to a function $\phi$ that describes the relative change of the plant population due to insect influence:

$$\phi(\mathbf{x}, \mathbf{a}, \mathbf{b}) = g_{\max} \rho_{\text{poll}}(\mathbf{x}, \mathbf{b}) - \rho_{\text{graz}}(\mathbf{x}, \mathbf{a}) = \frac{g_{\max}}{1 + \left(\sum_{i=1}^{n} b_i x_i\right)^{-1}} - \sum_{i=1}^{n} a_i x_i \tag{9}$$

where $g_{\max}$ is the growth rate of the plant population in the absence of death and with

## Recipe

### Checking Clone-Consistency

1. Extract relative growth rate.
   (Ignore immigration and similar external effects.)

2. Extract sums featuring any $x_i$.

3. Are they of the form $\sum_i \zeta(a_i)x_i$ with $\zeta(0) = 0$?

4. If not, cast them into this form by choosing parameters to be zero.

5. Are these choices biologically justified? (Depends on application.)

6. If 3 or 5 was positive for all sums, the model is clone-consistent.

## Example

$$\dot{x}_j = x_j g_j\left(c_j - x_j - \sum_{i \neq j} a_{ji}x_i\right)\left(1 - \frac{x_j}{1 + \sum_i b_{ji}^2 x_i}\right) + q_{\text{immigration}}$$

$$g_j\left(c_j - x_j - \sum_{i \neq j} a_{ji}x_i\right)\left(1 - \frac{x_j}{1 + \sum_i b_{ji}^2 x_i}\right)$$

$$x_j + \sum_{i \neq j} a_{ji}x_i \qquad \sum_i b_{ji}^2 x_i \qquad x_j$$

no **yes** no

$$\sum_i a_{ji}x_i \text{ with } a_{jj} = 1 \qquad \sum_i d_{ji}x_i \text{ with } d_{ji} = \begin{cases} 1 & \text{if } i = j \\ 0 & \text{if } i \neq j \end{cases}$$

**yes** **no**
(arbitrary example) (arbitrary example)

clone-inconsistent

### Building a Clone-Consistent Model

1. For what impacts shall the model account († on the focal population)?

2. Describe these in the form $\chi(\sum_i \zeta(a_i)x_i)$ with $\zeta(0) = 0$.

3. How do the impacts accumulate into the target quantity († relative growth $\phi$)?

4. † Insert into full model: $\dot{x}_j = x_j \phi$ or $x_j(t + \Delta t) = x_j(t) + \phi x_j(t)$ or similar.

 † This part only applies to models of population growth as in Eq. 10 (as opposed to, e.g., models of resource consumption).

flowering plants pollinated and grazed by insects

pollination rate loss to grazing

$$\rho_{\text{poll}}(\mathbf{x}, \mathbf{b}) = \frac{1}{1 + (\sum_i b_i x_i)^{-1}} \qquad \rho_{\text{graz}}(\mathbf{x}, \mathbf{a}) = \sum_i a_i x_i$$

$$\phi(\mathbf{x}, \mathbf{a}, \mathbf{b}) = g_{\max}\rho_{\text{poll}}(\mathbf{x}, \mathbf{b}) - \rho_{\text{graz}}(\mathbf{x}, \mathbf{a}) = \frac{g_{\max}}{1 + (\sum_i b_i x_i)^{-1}} - \sum_i a_i x_i$$

$$\dot{x}_j = x_j \phi(\mathbf{x}, \mathbf{a}, \mathbf{b}) = x_j\left(\frac{g_{\max}}{1 + (\sum_i a_i x_i)^{-1}} - \sum_i b_i x_i\right)$$

**Fig 3. Recipes for building and validating models using our framework.** In general $x_i$ denotes the abundance of population $i$, and $a_i$, $b_i$, $c_i$, and $d_i$ are parameters describing its impact. The example for checking is based upon Eq 21 and tailored for covering relevant cases. The example for building extends the one from *The form of impact functions*. For both examples, the biological background is discussed in more detail in the main text.

maximum fertilization. As $\phi$ is built from impact functions, it is an impact function itself. We can also see directly that $\phi$ complies with the general form of an impact function (Eq 6) by choosing $\chi(y, z) := \frac{g}{1+y^{-1}} - z$, $\omega_1 = \tau$, and $\omega_2 = \rho_{\mathrm{graz}}$. As a result, we can be certain that the resulting model satisfies the fundamental consistency criteria. Similarly, the basic building blocks we identified enable constructing consistent models of any other ecosystem.

**Assessing and building models.** Our treatment of impact functions allows us to ensure clone consistency when modeling the effects **of** a community. To ensure clone consistency of an entire population model, we additionally need to require clone consistency when modeling the effects **on** the growth of a population. Specifically, we require that the total growth of two populations of identical individuals must be the same as if those two populations were joined in the model.

We consider models of the forms:

$$\dot{x}_j = \mathcal{R}_j \qquad \text{or} \qquad x_j(t+1) = x_j(t) + \mathcal{R}_j, \tag{10}$$

where the right-hand side $\mathcal{R}_j$ describes the change of abundance of population $j$ due to the internal dynamics of the ecosystem. For simplicity, we omit the dependencies of $\mathcal{R}_j$ and ignore immigration and other external effects as well as dynamic variables that are not populations (such as explicitly modelled resources); adding them to such a model is straightforward.

We proved that $\mathcal{R}_j$ must have the form (see Methods, *Non-impact-function contribution to abundance changes must be proportional*):

$$\mathcal{R}_j = x_j \phi(\mathbf{x}, \mathbf{a}), \tag{11}$$

where $\phi$ is an impact function. This means that all dependencies of $\mathcal{R}_j$ on $\mathbf{x}$ must either happen within an impact function or in the form of a single factor $x_j$. Intuitively, each individual multiplies with a rate that is the result of all impacts it experiences in the ecosystem ($\phi$) – these impacts include interactions between individuals of the same population, e.g. due to crowding.

This insight provides an easy way to verify if models comply with our consistency criteria. We simply need to check if they have the form of Eq 11. To verify in turn if $\phi$ is an impact function, we can look for terms of the form of Eq 5. For instance, a common formulation of the generalized Lotka–Volterra model [1] can be rewritten as:

$$\dot{x}_j = g_j x_j \left( c_j - x_j - \sum_{i \neq j} a_{ji} x_i \right) = g_j x_j \left( c_j - \sum_i a_{ji} x_i \right), \tag{12}$$

with $a_{jj} = -1$, which reflects that a population maximally competes with itself. With this, the model is clearly built from a linear combination and a factor $x_j$ and we can thus be sure that the model is clone-consistent. We can now have another look at our introductory example (Eq 1): As the interaction term features logarithms of abundances, it does not comply with the form of Eq 5. Thus, the model violates our consistency criteria and the observed clone inconsistency (Fig 1) is inevitable. We summarize the recipe for checking a model and provide a more extensive example in the top of Fig 3.

This recipe can be inverted to build a clone-consistent model. In the example from *The form of impact functions*, we can directly insert Eq 9 into Eq 11 and obtain a model for the change of a plant population in light of pollination and grazing (see Fig 3 bottom). If we stringently apply our framework here, the only thing that distinguishes plants and insects is that the former have a grazing and pollination rate of zero.

Our framework can also be applied to experiments that do not assess the details of ecological interactions (nutrients, toxins, etc.) but only aggregated, phenomenological *interaction*

*observables*, such as the carrying capacity of a population in the presence of another. This is typical for high-throughput experiments assessing the pair-wise interactions of microbial communities [15, 16, 19–22]. Combining Eqs 10, 11, 6, and 5, we obtain a general ansatz for such a model:

$$\dot{x}_j = x_j \chi \left( \sum_{i=1}^{n} a_{ji1} x_i, \sum_{i=1}^{n} a_{ji2} x_i, \ldots, \sum_{i=1}^{n} a_{jim} x_i \right). \tag{13}$$

Here, we refrain from transforming the parameters (corresponding to $\zeta$ = id in Eq 5), since this would not affect the final model. First we identify the number of basic impact functions $m$ as the number of interaction observables: Using fewer than $m$ basic impact functions would entail that interaction observables would go unused, i.e. available information on the system would be ignored; more than $m$ basic impact functions would make the model overly complex given our limited knowledge of the system. The choice of the combining function $\chi$ depends on the application, but we expect that a product of transformations of individual basic impact functions is often appropriate. In this case our ansatz becomes:

$$\dot{x}_j = x_j \prod_{k=1}^{m} \eta_k \left( \sum_{i=1}^{n} a_{jik} x_i \right), \tag{14}$$

with some transformation functions $\eta_k$. With $\chi$ chosen, the parameters and functions ($a_{jik}$ and $\eta_k$) can be determined using:

- the requirement that the model should reproduce key characteristics of the experimental scenarios, e.g., the carrying capacity of a population,

- ecological assumptions and facts about the scenario, e.g., that the predation rate should increase with the abundance of predators, or

- assumptions of simplicity (Occam's razor), e.g., that $\eta_k$ should not be more complex than necessary to fulfill the experimental and ecological constraints.

In *New model*, we provide an example for this approach.

**Higher-order interactions.** Our framework is readily extended to models describing second- or higher-order interactions. For this, one simply has to consider parameters that are associated with more than one population. Then, it is convenient to use other basic building blocks (instead of Eq 5), for example for second-order interactions:

$$\mathbf{x}, \mathbf{a} \mapsto \sum_{i=1}^{n} \sum_{j=1}^{n} \zeta(a_{ij}) x_i x_j, \tag{15}$$

which is an impact function independent of whether $a_{ij}$ is considered a parameter associated with population $i$ or with population $j$. Analogous building blocks exist for higher interaction orders. Such building blocks are featured in existing models that capture higher-order interactions [12, 13, 17].

## Case study: Semi-empirical models for microbial communities

As an instructive example, we apply the impact-function framework to describe the dynamics of a microbial community for which multiple ecological interaction observables were recently measured experimentally. We first describe the experiment and the general modeling challenge. Then we show that an existing model is clone-inconsistent and discuss the specific

**Fig 4. Acquisition of data used in our case study.** Measurement of growth characteristics (left) and pairwise interactions (right) of bacterial strains isolated from urinary-tract infections [16]. Left: Each strain $j$ was cultivated for 48 h in artificial urine. Solid bold letters represent individuals of the respective strain. The exponential growth rate $g_j$ as well as the carrying capacity $c_j$ (named *yield* in the original study [16]) were experimentally determined via optical densities. Right: For each strain $k$, a conditioned medium was produced by letting the strain grow for 48 h, mechanically removing the bacteria to obtain a *supernatant*, and mixing the result with fresh medium in a ratio of $v := 0.4$. Outline letters ("footprints") indicate to what extent the respective culture consists of supernatant. In each such medium, each strain $j$ was cultivated, and the conditioned growth rate $g_{jk}$ and carrying capacity $c_{jk}$ were determined as above.

reasons and implications of this. Finally, we show how our framework can be used to build a new clone-consistent model.

A recent study (co-authored by one of us) used a high-throughput approach to systematically measure ecological interactions in microbial communities consisting of strains isolated from polymicrobial urinary-tract infections (UTI) [16]. For each strain, the exponential growth rate $g_j$ and the carrying capacity $c_j$ in isolation were measured (Fig 4, left). For convenience, abundances of each strain $j$ were normalized such that $c_j = 1$. Furthermore, for each strain $k$, a medium partially conditioned by that strain was produced (it contains a fraction $v$ of the supernatant). In each such partially conditioned medium, the conditioned growth rate $g_{jk}$ and carrying capacity $c_{jk}$ of each strain $j$ were measured to quantify how strain $k$ affects strain $j$ (Fig 4, right). Creating a model from this dataset is particularly challenging since it has to include two interaction observables: growth rate and carrying capacity in conditioned medium.

For modeling this system, it is a plausible assumption that the abundance of a population also represents its footprint, i.e. the nutrients, toxins, and other relevant substances produced or depleted by that population. The basis for this simplifying assumption is that populations are declining only due to dilution of the entire system, which has the same effect on the footprint. With this assumption, we can treat the medium partially conditioned by strain $k$ as an ecosystem where the abundance of that strain is fixed to the corresponding fraction of its carrying capacity ($x_k = v c_k = v$).

The general form of an ordinary differential equation describing the (normalized) abundance $x_j$ of population $j$ in this system is $\dot{x}_j = \mathcal{R}_j(\mathbf{x}, \mathbf{a})$. Such models should reproduce the observed growth rates and carrying capacities for all situations that were experimentally investigated. For instance, in the absence of other strains, the initial exponential growth rate of strain $j$ in the model should be equal to its experimentally observed exponential growth rate $g_j$:

$$\lim_{x_j \to 0} \frac{\mathcal{R}_j(\vec{0}, \mathbf{a}, \mathrm{x}_j = x_j)}{x_j} = \frac{\partial \mathcal{R}_j}{\partial \mathrm{x}_j}(\vec{0}, \mathbf{a}) = g_j, \tag{16}$$

where the argument $(\vec{0}, \mathbf{a}, \mathrm{x}_j = x_j)$ of $\mathcal{R}_j$ denotes that the abundance of population $j$ is $x_j$ and all other abundances are zero. Similarly, we can deduce three other such criteria, resulting in

one criterion per experimental observable:

$$\text{carrying capacity:} \quad \mathcal{R}_j(\vec{0}, \mathbf{a}, \mathrm{x}_j = 1) = 0, \tag{17}$$

$$\text{conditioned growth rate:} \quad \frac{\partial \mathcal{R}_j}{\partial \mathrm{x}_j}(\vec{0}, \mathbf{a}, \mathrm{x}_k = v) = g_{jk}, \tag{18}$$

$$\text{conditioned carrying capacity:} \quad \mathcal{R}_j(\vec{0}, \mathbf{a}, \mathrm{x}_j = c_{jk}, \mathrm{x}_k = v) = 0. \tag{19}$$

**Existing model.** The original study [16] proposed a model for communities consisting of such strains based on Verhulst's logistic model for one population [65]:

$$\dot{x} = xg\left(1 - \frac{x}{c}\right), \tag{20}$$

where both the growth rate $g$ and the carrying capacity $c$ (normalized to 1 here) are modified by interaction terms incorporating the experimentally obtained parameters:

$$\dot{x}_j = \underbrace{x_j g_j \left[1 + \sum_{i \neq j} a_{ji} x_i\right]}_{\text{growth term}} \underbrace{\left(1 - \frac{x_j}{\lceil 1 + \sum_{i \neq j} b_{ji} x_i \rceil}\right)}_{\text{carrying-capacity term}}, \tag{21}$$

with $\lceil z \rceil := \max(0, z)$, $a_{ji} := \frac{g_{ji}}{g_j} - 1$, $b_{ji} := \begin{cases} c_{ji} - 1 & \text{if } c_{ji} \geq 1 \\ \frac{1}{v}\left(c_{ji} - 1\right) & \text{if } c_{ji} < 1 \end{cases}$, and the remaining symbols

are as in Fig 4. Looking at this model through the lens provided by our framework, both the carrying-capacity and the growth term should be impact functions, with $a$ and $b$ being the parameters quantifying these impacts. However, it is clear that neither term is built from linear combinations (with complete sums), matching the form of Eq 5. Thus the model must violate at least one of our criteria. As it clearly satisfies Criteria I1–I4, it therefore must violate I4 and be clone-inconsistent which can indeed be shown explicitly (S1 Appendix). We can complete the sums with appropriate choices of $a_{jj}$ and $b_{jj}$, and we can expand the solitary $x_j$ in the numerator to:

$$x_j = \sum_i d_{ji} x_i \quad \text{with} \quad d_{ji} = \begin{cases} 1 & \text{if } i = j \\ 0 & \text{if } i \neq j \end{cases}, \tag{22}$$

where the new parameter $d_{ji}$ can be interpreted as quantifying the extent to which population $i$ occupies the niche of population $j$. However, this expansion implicitly assumes that the strains have non-overlapping niches, which is not justified for this study, as it features many communities containing two strains of the same genus or even species. This interpretation of $d_{ji}$ would also ignore that the conditioned carrying capacities $c_{ji}$ capture niche overlap. Following the recipe from Fig 3 (top), Question 5 must be answered with *no* at this point.

The fixed points of this model are characterized by:

$$x_j = 1 + \sum_{i \neq j} b_{ji} x_i \quad \Leftrightarrow \quad 0 = 1 + \sum_i b_{ji} x_i \text{ with } b_{jj} = -1. \tag{23}$$

Consequently, they are clone-consistent (also see S1 Appendix). Most conclusions of the original study [16] are based on these fixed points and thus unaffected by the clone

inconsistency of the model. However, clone inconsistency affects the transient dynamics (see also S1 Appendix), which is relevant as these communities are subject to frequent dilutions (due to bladder voiding) which can happen long before the system has equilibrated.

**New model.** We use our framework to construct a new population dynamics model for this scenario. Since two experimental interaction observables are available, we make an ansatz using two basic impact functions (see *Assessing and building models*). As one of the observables and thus one of the impact functions captures the carrying capacity, we choose to multiply (and not add) the impact functions to ensure that an impact function can single-handedly reduce the growth to zero. Our ansatz is thus Eq 14 with $m = 2$:

$$\dot{x}_j = \mathcal{R}_j(\mathbf{x}, \mathbf{a}) \coloneqq x_j \rho_j \left( \sum_{i=1}^{n} r_{ji} x_i \right) \varsigma_j \left( \sum_{i=1}^{n} s_{ji} x_i \right), \tag{24}$$

where $a_{j1} \coloneqq r_j$ and $a_{j2} \coloneqq s_j$.

Inserting this ansatz into Eqs 16–19 and making a few choices that do not affect generality already yields strong constraints on the functions $\rho_j$ and $\varsigma_j$ and on how the parameters $r_{jk}$ and $s_{jk}$ relate to these and the experimental parameters $g_j$, $g_{jk}$ and $c_{jk}$, namely (see Methods, *Deriving a new model for UTI strains – the legwork*):

$$\varsigma_j(0) = 1, \quad \varsigma_j(1) = 0, \qquad s_{jk} = \frac{1 - c_{jk}}{v}, \qquad \rho_j(0) = g_j, \quad \rho(r_{jk}v) = \frac{g_{jk}}{\varsigma(1 - c_{jk})}. \tag{25}$$

By further making simple choices for $\rho_j$ and $\varsigma_j$ within these constraints and accounting for singularities and discontinuities (see Methods, *Deriving a new model for UTI strains – the legwork*), we arrive at the following model (with $\lceil z \rceil \coloneqq \max(0, z)$):

$$\dot{x}_j = x_j \underbrace{\left[ g_j + \sum_{i=1}^{n} \frac{g_{ji} - g_j}{v} x_i \right]}_{\text{growth term}} \cdot \underbrace{\left[ 1 - \left\lceil \sum_{i=1}^{n} \frac{1 - c_{ji}}{v} x_i \right\rceil^q \right]}_{\text{carrying-capacity term}}. \tag{26}$$

Like the existing model (Eq 21), this can be understood as an expansion of Verhulst's logistic model with the following differences: First and foremost, the carrying-capacity term is simplified to achieve clone consistency – interactions affecting the carrying capacity of population $j$ are now captured in a single sum. Second, the additional parameter $q$ governs how abruptly the saturation effect kicks in. Third, the dilution factor $v$ is included consistently without case distinctions. Fourth, as per the initial assumptions, populations cannot decline anymore (unless dilution is added to the model). The particular form of this model illustrates how challenging it can be to write down clone-consistent models from scratch without using the framework presented here.

We find that this model can explain observed species abundances and ecological stability in a small experimental dataset (S1 Appendix) at least as well as the previous model (Eq 21). Moreover, the fixed points of both models are the same under most conditions (S1 Appendix).

## Implications

**Specific implications: Clone-inconsistency points to implicit assumptions.** While many popular models, including most variants of the generalized Lotka–Volterra model [1], comply with our criteria, others do not [50, 52: Eqs 9 and 10, 53, 54: Eqs 1.28–1.30 and 1.50, 55: Eqs 11 and 12, 56, 57, 58: Eq 5, 59: NFR model, 60, 16, 61, 62: Eq 3, 3: Figs 3b and c, 63: UIM and IIM model]. However, as we will elaborate below, this does not mean that these models should be dismissed outright.

Any model can be made clone-consistent by setting the right parameters to zero (Fig 3 and Eq 22). For instance, suppose that $u - x_j$ quantifies the unoccupied portion of the niche of population $j$. We can extend this term to an impact function:

$$u - x_j = u - \sum_i a_{ji} x_i, \tag{27}$$

with $a_{ii} = 1$ and $a_{ji} = 0$ for $j \neq i$. Here $a_{ji}$ describes the extent to which population $i$ occupies the niche of population $j$. Thus, $a_{ji} = 0$ for $j \neq i$ implies that population $j$ exclusively occupies its niche. Whether this assumption of an exclusive niche is justified depends on the specific application. The strength of our framework is to systemically point to such (often implicit) assumptions and prompt an explicit justification or an improvement of the model if the assumptions are not justified.

As an instructive example, consider a recently proposed unique-interactions model (UIM) [63]. This model features three interaction terms. Two describe the influence of mutualistic or exploitative interactions, respectively, on a focal population $j$ as:

$$\sum_i \frac{a_{ji} x_i}{h + x_i}, \tag{28}$$

where $a_{ji}$ describes the strength of the respective interaction and $h$ is the half-saturation constant. These terms are not impact functions and thus clone-inconsistent. However, a key assumption of this model is that mutualistic and exploitative interactions are unique for each focal population, which is a legitimate approximation for the purposes of that work. Following this assumption, each summand of Eq 28 represents a separate interaction mechanism, such as a specific resource, service, or mode of predation, and only population $i$ affects the focal population $j$ via this mechanism. In our framework, each summand can be expanded to an impact function, with only one parameter different from zero. By contrast the third term in the UIM model, which describes competitive interactions, is clone-consistent. For this term, unique interactions are neither assumed nor would this be justified.

A similar case of assumptions justifying clone inconsistency arises if some features of populations cannot be feasibly encoded in numerical parameters, which we assumed as given so far. For example, compatibility for sexual reproduction is tedious to capture in a parameter; instead, it is often reasonable to assume that populations do not interbreed (i.e., contain species as defined by Mayr). In this case, splitting a population in two equal parts also halves the availability of partners for sexual reproduction. Thus, this availability should not be described by a clone-consistent impact function. These examples illustrates how clone-inconsistent terms can make sense in a model if properly justified.

Many of the aforementioned studies based on clone-inconsistent models primarily make statements about the effects of model properties on population dynamics. If these findings are based on strong assumptions, their generality, relevance, and applicability are considerably diminished. Moreover, if these assumptions are implicit, this increases the risk that the study is misinterpreted and misapplied by others. The framework presented here will help to avoid these problems by forcing key assumptions of ecological models to be explicitly stated and justified.

**Scope of the framework.**   Our approach extends to diverse types of models. In particular, it is not restricted to models employing ordinary differential equations, but also applicable to models with noise, time delays, or discrete time steps. Further, higher-order interactions are covered by our framework. We mainly used impact functions to describe the impact of a community on a population. However, both the target and the source can be other entities, e.g. the availability of a resource, the concentration of a toxin, or an aggregated observable such as the albedo of foliage or the pH value of a growth medium. A common case is the impact of the

community on a resource within a consumer–resource model [66, 67]. Going beyond modeling, impact functions that describe observables are closely linked to the requirement for ecological observables – such as diversity – to be clone-consistent [39–41].

Notably, the framework presented here has conceptual parallels to pharmacological approaches that are widely used as null models for the combined effect of two drugs [68, 69]: One is *Loewe additivity*, which is based on arguments similar to clone consistency and is a suitable reference if the two drugs target the same component of the cell. The other is *Bliss independence*, which violates clone consistency at first glance and is suitable if the two drugs affect different components of the cell. In our framework, drugs that target the same cell component correspond to using the same interaction mechanism and thus would be captured by the same basic impact function. The effect of a complex drug cocktail could be captured by several Bliss-independent basic impact functions, each of which comprises a series of Loewe-additive components.

**General implications.**    While each instance of unjustified clone inconsistency reflects a shortcoming of the model, general statements about the consequences of clone inconsistency for the solutions of a model are most likely impossible. Clone inconsistency is tightly intertwined with the fabric of the model, and thus we cannot study its effect in isolation. Moreover, there is no reason to expect that clone-inconsistent models have any relevant commonalities, as they are extremely diverse. For example, it makes a difference whether joining identical populations increases or decreases some impact (that would be unchanged in a clone-consistent model). For illustration, the diversity of clone-inconsistent models may be compared to that of all numbers not divisible by seven. Similarly, it is likely not possible to draw general conclusions about the dynamic behavior of the clone-consistent models – such as favoring or suppressing oscillatory dynamics. While they are based on linear combinations, they can certainly be non-linear. Not only can a non-linear function still be applied to the linear combination (e.g., $\chi$ in Eq 6), but any existing non-linear model can be made clone-consistent by sufficiently strong assumptions. The central question is whether these assumptions are biologically justified. However, insights can be gained by interpreting each impact function as a mechanism of ecological interaction:

Many pure modeling studies use a model of the general form [12, 59, 62]:

$$\dot{x}_j = x_j\left(g_j + \sum_{i=1}^{n}\eta_{ji}(x_i)\right),\tag{29}$$

where $g_j$ is the unperturbed growth rate of population $j$. If all $\eta_{ji}$ are linear, this kind of model uses a single basic impact function and requires no further assumptions. However, within our framework, non-linear $\eta_{ji}$ can only be justified as follows: Each summand in Eq 29 corresponds to one interaction mechanism by which population $i$ uniquely affects population $j$. In this case, each summand would be expanded to an impact function, where all parameters except one are zero. This would result in a total of $n$ basic impact functions. A notable implication is that models of the above form can either feature 1 or $n$ basic impact functions (depending on whether $\eta_{ji}$ is linear), but they cannot capture the middle ground in between. As each basic impact function can be associated with one interaction mechanism, this limitation is relevant beyond the consistency issues addressed by our framework.

To fill this gap, our framework suggests an alternative form for general ecosystem models, such as:

$$\dot{x}_j = x_j g_j \prod_{k=1}^{m}\eta_k\left(\sum_{i=1}^{n}a_{jik}x_i\right),\tag{30}$$

where $m$ is the number of impact functions (compare to Eq 14). Each factor in Eq 30 corresponds to one interaction mechanism, which can involve multiple populations. To capture

that most populations do not participate in most interaction mechanisms, we propose to sample the interaction coefficients ($a_{ijk}$) from a distribution containing many zeros. By narrowing down or expanding the space of possible models taken into consideration, the alternative form for general models above can inform studies employing random interaction parameters [9–13], generalized modeling [25], or machine learning.

## Conclusion

We introduced a framework for building ecosystem models using impact functions as building blocks. This framework is aimed at ensuring the clone consistency of models and thus constrains the possible choices of models. While at first this may seem like at a burden, we anticipate that it will rather facilitate the modeling process by guiding ecologists when choosing from the (still infinitely many) clone-consistent models. Our framework further prompts relevant questions about the underlying assumptions of models. Specifically, the absence of impact functions in a model exposes that it is clone-inconsistent, which may indicate a fundamental problem. Alternatively, clone-inconsistency can reveal implicit assumptions at the heart of the model, which need to be justified and may limit the model's generality. Our framework also informs the form of more general models by outlining the space of possible models for ecosystems and enables new research directions in this field. Finally, our approach could be extended to implement criteria for specific ecological scenarios such as predation [27–30, 33]. Overall, the framework presented here provides a systematic way to understand models and can form the backbone for a wide range of ecological modeling studies.

## Methods: Mathematical backbone

### Mathematical notation

- **lowercase italic letters**: numbers or parameter configurations (tuples of numbers);

- **lowercase Greek letters**: functions;

- **boldface letters**: vectors or similar;

- **uppercase letters**: sets of respective contents;

- $n$: the number of populations;

- $m$: the number of parameters per population;

- $\mathbb{R}_+$: the non-negative real numbers;

- $X = \mathbb{R}_+^n$: the space of all possible population abundances;

- $A = \mathbb{R}^{n \times m}$: the space of all possible parameter configurations of these populations;

- $X \times A$: the domain of impact functions;

- $\mathbf{x} = (x_1, \ldots, x_n) \in X$: an arbitrary first argument of an impact function (abundances), where $x_i$ is the abundance of population $i$;

- $\mathbf{a} = (a_1, \ldots, a_n) \in A$: an arbitrary second argument of an impact function (parameters), where $a_i \in \mathbb{R}^m$ are the parameter values that describe population $i$;

- **non-italic sans-serif letters**: modifications of specific components of arguments of an impact function (similar to named arguments in many programming languages). For example: $\phi(\mathbf{x}, \mathbf{a}, \mathsf{x}_2 = y)$ denotes $\phi((x_1, y, x_3, \ldots, x_n), (a_1, \ldots, a_n))$. Here the arguments of the function $\varphi$ are

**x** and **a** except for the abundance of the second population ($x_2$) being changed to $y$.

- $\mathcal{Y} \mapsto \mathcal{Z}$: the function that maps $\mathcal{Y}$ to $\mathcal{Z}$ (anonymous function).

- $\mathcal{Y} := \mathcal{Z}$: $\mathcal{Y}$ is defined as $\mathcal{Z}$.

*Proof*: $\Xi$ *generates* $\Phi$ requires additional notation that is mostly introduced in that subsection.

## The functional algebra of impact functions

In this subsection, we describe our main mathematical result and connect it to the main text, from encoding our criteria mathematically to translating the results back to application.

Expressed in equations, our criteria for impact functions are:

**I1** Commutativity:

$$\forall\, i \neq j: \ \phi(\mathbf{x}, \mathbf{a}) = \phi(\mathbf{x}, \mathbf{a}, x_i = x_j, x_j = x_i, a_i = a_j, a_j = a_i). \tag{31}$$

**I2** When a population is absent, its associated parameters have no effect:

$$\forall\, a_1, b_1 \in \mathbb{R}^m: \ \phi(\mathbf{x}, \mathbf{a}, x_1 = 0, a_1 = a_1) = \phi(\mathbf{x}, \mathbf{a}, x_1 = 0, a_1 = b_1). \tag{32}$$

**I3** When all parameters associated with a given population are zero, that population has no impact:

$$\forall\, x_1, y_1 \in \mathbb{R}_+: \ \phi(\mathbf{x}, \mathbf{a}, x_1 = x_1, a_1 = 0) = \phi(\mathbf{x}, \mathbf{a}, x_1 = y_1, a_1 = 0). \tag{33}$$

Note that the parameter value corresponding to no impact could be readily changed from zero to any other value.

**I4** Clone consistency:

$$\forall\, z \in [-x_1, x_2]: \tag{34}$$
$$\phi(\mathbf{x}, \mathbf{a}, a_1 = a_2 = b) = \phi(\mathbf{x}, \mathbf{a}, x_1 = x_1 + z, x_2 = x_2 - z, a_1 = a_2 = b).$$

Note that through commutativity (I1), the other criteria apply to all populations or pairs of populations of the impact function $\phi$, respectively (and not just to populations 1 and 2). Clone consistency (I4) of more than two populations is covered by applying the respective criterion repeatedly.

In the terms of functional analysis, impact functions form a *functional algebra* $\Phi$. This means that each product or sum of two impact functions is again an impact function and that each multiple of an impact function is an impact function. This algebra is also closed, which means that the limit of uniformly converging sequences of impact functions is again an impact function.

To easily build and detect impact functions, it is crucial to find a (small) set of impact functions from which all impact functions can be build, i.e., a *generating set* of $\Phi$. Our main mathematical result is that $\Xi = \Lambda \cup \Gamma$ is such a generating set, where $\Gamma$ is the set of constant functions and taking the limit of a uniformly converging sequence is considered amongst the generating operations. $\Lambda$ is the set of all linear combinations of powers of parameters and abundances, i.e., functions of the form:

$$\mathbf{x}, \mathbf{a} \mapsto \sum_{i=1}^{n} a_{ik}^p x_i \tag{35}$$

for some $p \in \{1, \ldots\}$ and for some $k \in \{1, \ldots, m\}$.

We will formally prove this in the next subsection (*Proof*: Ξ *generates* Φ), but the essential idea is this: Our criteria (I1–I4) require an impact function to have the same value on given subsets of its domain. For Ξ to be a generating set of Φ, it must reflect this: First, the functions in Ξ must be constant on each such subset, i.e., fulfill our criteria for impact functions. Otherwise Ξ would also generate functions that are not impact functions. (It is straightforward to show that the elements of Ξ fulfill our criteria, however, in our proof this is a byproduct.) Second and more crucially, for each pair of points that are not in the same subset, there must be a function in Ξ that differs between these points (this is called *separating points*). Otherwise some impact functions could not be generated by Ξ. We show the latter by using our criteria (I1–I4) to systematically transform arguments of impact functions to a *canonical form*, in which populations are ordered by impact and maximally lumped together.

In application, the fact that Φ is a closed functional algebra, i.e., that limits remain within it, is relevant as it addresses the case of a (non-polynomial, continuous) function being applied to the result of an entire impact function or the parameters. We can rewrite:

$$\sum_{p=1}^{\infty} v_p \sum_{i=1}^{n} a_i^p x_i = \sum_{i=1}^{n} \left( \sum_{p=1}^{\infty} v_p a_i^p \right) x_i = \sum_{i=1}^{n} \zeta(a_i) x_i, \tag{36}$$

with $\zeta(z) = \sum_{p=1}^{\infty} v_p z^p$. This allows us to use building blocks of the form of Eq 5 instead of Eq 35. We can also rewrite:

$$\sum_{q=1}^{\infty} u_q \left( \sum_{i=1}^{n} a_i x_i \right)^q = \chi \left( \sum_{i=1}^{n} a_i x_i \right), \tag{37}$$

with $\chi(z) = \sum_{q=1}^{\infty} u_q z^q$. This allows for "wrapping" functions around impact functions.

## Proof: Ξ generates Φ

We here state and prove our main mathematical result, namely:

**Theorem 1**. *Let* $\Lambda_k := \{ \mathbf{x}, \mathbf{a} \mapsto \sum_{i=1}^{n} a_{ik}^p x_i | p \in \{1, \ldots\} \}$ *denote the set of linear combinations of powers of values of the k-th parameter and abundances. Denote the set of all such functions as* $\Lambda = \bigcup_{k=1}^{m} \Lambda_k$. *Let* $\Xi := \Lambda \cup \Gamma$, *where* $\Gamma$ *is the set of constant functions. Let* Ψ *be the generated set of* Ξ, *i.e., the smallest closed functional algebra that contains* Ξ. *Then* Ψ = Φ, *i.e.,* Φ *contains all impact functions as characterized by criteria I1–I4.*

To prove it, we apply Bishop's Theorem [70, 71]. We here only need the reduction to the special case of real-valued functions (as opposed to complex-valued functions):

**Bishop's Theorem**. *Let Z be a compact Hausdorff space. Let* Ψ *be a closed unital subalgebra of* $C(Z, \mathbb{R})$. *Let* $\phi \in C(Z, \mathbb{R})$. *Suppose that* $\phi|_S$ *is constant for each subset* $S \in Z$ *such that* $\psi|_S$ *is constant for all* $\psi \in \Psi$. *Then* $\phi \in \Psi$.

The requirements of Bishop's Theorem on Ψ are fulfilled since $Z$ can be any sufficiently large compact subset of $X \times A$ and the inclusion of Γ ensures unitality. To show that the functional algebra Ψ contains all impact functions, we therefore need to show that for an arbitrary impact function $\phi$ for any $\mathbf{x}, \hat{\mathbf{x}} \in X$ and $\mathbf{a}, \hat{\mathbf{a}} \in A$:

$$\forall \psi \in \Psi : \psi(\mathbf{x}, \mathbf{a}) = \psi(\hat{\mathbf{x}}, \hat{\mathbf{a}}) \quad \Rightarrow \quad \phi(\mathbf{x}, \mathbf{a}) = \phi(\hat{\mathbf{x}}, \hat{\mathbf{a}}), \tag{38}$$

or, in the language of functional analysis, Ψ has to *separate points*, except when no impact function separates those points. Since point-separations are unaffected by algebraic

operations of functions and limits, $\Psi$ separates points, if and only if $\Xi$ does. Moreover, since the functions from $\Gamma$ are constant everywhere (and thus separate no points at all), this is equivalent to $\Lambda$ separating points. Finally, since for any $i \neq j$, the functions from $\Lambda_i$ are constant wherever the functions from $\Lambda_j$ are not, it suffices to only consider one $\Lambda_i$, i.e., scalar parameters ($m = 1$).

We prove that $\Lambda$ separates points for $m = 1$ with three lemmas, for which we transform the arguments to a canonical form (Definitions 1–4), in which populations are ordered by impact and maximally lumped together. We first show that if all functions from $\Lambda$ have the same value for two arguments, these arguments have the same canonical form (Lemma 1). We then employ our criteria to show that no impact function will differ for two arguments that have the same canonical form (Lemma 2). Finally, we combine the first two lemmas to show that if all functions from $\Lambda$ have the same value for two arguments, so do all impact functions (Lemma 3). Thus $\Lambda$ separates points where it needs to (as per Eq 38).

**Definition 1**. *Let* $\mathbf{x} \in X$ *and* $\mathbf{a} \in A$. *Let* $\mathfrak{A} = (\mathfrak{a}_1, \ldots, \mathfrak{a}_s)$ *be the ordered sequence of non-zero values of* $\mathbf{a}$ *that correspond to a non-zero abundance, i.e.:*

$$\mathfrak{a}_1 < \mathfrak{a}_2 < \ldots < \mathfrak{a}_s \text{ and } \forall \mathfrak{a} \in \mathfrak{A} : \exists i \in \{1, \ldots, n\} : \mathfrak{a} = a_i \neq 0 \wedge x_i > 0. \tag{39}$$

As the $\mathfrak{a}$ are unique and ordered, we will directly use them like indices to avoid additional levels of indexing. One can think of them as equivalence classes of parameters.

**Definition 2**. *Let* $\mathbf{x} \in X$ *and* $\mathbf{a} \in A$. *For a given* $\mathfrak{a}$, *let* $I_\mathfrak{a}$ *be the set of indices where this parameter value is assumed and the corresponding abundance is not zero, i.e., the maximal set $I$ such that* $a_i = \mathfrak{a}$ *and* $x_i > 0$ *for all* $i \in I_\mathfrak{a}$. *Consequentially,* $I_\mathfrak{b} := \{\}$ *for* $\mathfrak{b} \notin \mathfrak{A}$.

**Definition 3**. *Let* $\mathbf{x} \in X$ *and* $\mathbf{a} \in A$. *Denote the sums of abundances for one absolute parameter value as* $z_\mathfrak{a} := \sum_{i \in I_\mathfrak{a}} x_i$.

**Lemma 1**. *Suppose* $\mathbf{x}, \hat{\mathbf{x}} \in X$ *and* $\mathbf{a}, \hat{\mathbf{a}} \in A$ *are such that*:

$$\forall p \in \{1, \ldots\} : \quad \sum_{i=1}^{n} a_i^p x_i = \sum_{i=1}^{n} \hat{a}_i^p \hat{x}_i. \tag{40}$$

*Then*:

$$\forall \mathfrak{a} \in \mathfrak{A} \cup \hat{\mathfrak{A}} : \quad z_\mathfrak{a} = \sum_{i \in I_\mathfrak{a}} x_i = \sum_{i \in \hat{I}_\mathfrak{a}} \hat{x}_i =: \hat{z}_\mathfrak{a}. \tag{41}$$

We show Eq 41 by induction over $\mathfrak{A} \cup \hat{\mathfrak{A}}$ in descending order of absolute value. We first note that the lemma trivially holds for all $\mathfrak{a} \in \{\}$. In the following we show that, if the lemma holds for all $\mathfrak{b}$ with $|\mathfrak{b}| > \mathfrak{a} > 0$, it also holds for $\mathfrak{a}$ and $-\mathfrak{a}$. (If one of $I_\mathfrak{a} \cup \hat{I}_\mathfrak{a}$ and $I_{-\mathfrak{a}} \cup \hat{I}_{-\mathfrak{a}}$ is empty, this does not affect this part of the proof.) To this end, we first show that the linear combinations must also be equal when only considering coefficients $\mathfrak{c}$ with $|\mathfrak{c}| \leq \mathfrak{a}$ (for all $p$):

$$\sum_{|\mathfrak{c}| \leq \mathfrak{a}} \sum_{i \in I_\mathfrak{c}} a_i^p x_i = \sum_{\mathfrak{d}} \sum_{i \in I_\mathfrak{d}} a_i^p x_i - \sum_{|\mathfrak{b}| > \mathfrak{a}} \sum_{i \in I_\mathfrak{b}} a_i^p x_i \stackrel{\text{D 2}}{=} \sum_{i=1}^{n} a_i^p x_i - \sum_{|\mathfrak{b}| > \mathfrak{a}} \mathfrak{b}^p \sum_{i \in I_\mathfrak{b}} x_i$$

$$\stackrel{\text{Eqs 40,41}}{=} \sum_{i=1}^{n} \hat{a}_i^p \hat{x}_i - \sum_{|\mathfrak{b}| > \mathfrak{a}} \mathfrak{b}^p \sum_{i \in \hat{I}_\mathfrak{b}} \hat{x}_i \stackrel{\text{D 2}}{=} \sum_{\mathfrak{d}} \sum_{i \in \hat{I}_\mathfrak{d}} \hat{a}_i^p \hat{x}_i - \sum_{|\mathfrak{b}| > \mathfrak{a}} \sum_{i \in \hat{I}_\mathfrak{b}} \hat{a}_i^p \hat{x}_i = \sum_{|\mathfrak{c}| \leq \mathfrak{a}} \sum_{i \in \hat{I}_\mathfrak{c}} \hat{a}_i^p \hat{x}_i$$

$$\tag{42}$$

If $z_\mathfrak{a} + z_{-\mathfrak{a}} \neq 0$ and $\hat{z}_\mathfrak{a} + \hat{z}_{-\mathfrak{a}} \neq 0$ the above equality will be dominated by $\mathfrak{a}^p$ for $p \to \infty$, which gives us:

$$
1 \overset{\text{Eq 42}}{=} \lim_{\substack{p \to \infty \\ p \text{ even}}} \frac{\sum_{|\mathfrak{c}| \leq \mathfrak{a}} \sum_{i \in I_\mathfrak{c}} a_i^p x_i}{\sum_{|\mathfrak{c}| \leq \mathfrak{a}} \sum_{i \in \hat{I}_\mathfrak{c}} \hat{a}_i^p \hat{x}_i} = \lim_{\substack{p \to \infty \\ p \text{ even}}} \frac{\sum_{i \in I_\mathfrak{a} \cup I_{-\mathfrak{a}}} a_i^p x_i}{\sum_{i \in \hat{I}_\mathfrak{a} \cup \hat{I}_{-\mathfrak{a}}} \hat{a}_i^p \hat{x}_i} \overset{\text{D 2}}{=} \lim_{\substack{p \to \infty \\ p \text{ even}}} \frac{\sum_{i \in I_\mathfrak{a}} \mathfrak{a}^p x_i + \sum_{i \in I_{-\mathfrak{a}}} (-\mathfrak{a})^p x_i}{\sum_{i \in \hat{I}_\mathfrak{a}} \mathfrak{a}^p \hat{x}_i + \sum_{i \in \hat{I}_{-\mathfrak{a}}} (-\mathfrak{a})^p \hat{x}_i}
$$

$$
= \lim_{\substack{p \to \infty \\ p \text{ even}}} \frac{\mathfrak{a}^p \left( \sum_{i \in I_\mathfrak{a}} x_i + \sum_{i \in I_{-\mathfrak{a}}} x_i \right)}{\mathfrak{a}^p \left( \sum_{i \in \hat{I}_\mathfrak{a}} \hat{x}_i + \sum_{i \in \hat{I}_{-\mathfrak{a}}} \hat{x}_i \right)} = \frac{\sum_{i \in I_\mathfrak{a}} x_i + \sum_{i \in I_{-\mathfrak{a}}} x_i}{\sum_{i \in \hat{I}_\mathfrak{a}} \hat{x}_i + \sum_{i \in \hat{I}_{-\mathfrak{a}}} \hat{x}_i} \overset{\text{D 3}}{=} \frac{z_\mathfrak{a} + z_{-\mathfrak{a}}}{\hat{z}_\mathfrak{a} + \hat{z}_{-\mathfrak{a}}} \tag{43}
$$

$$
\Rightarrow z_\mathfrak{a} + z_{-\mathfrak{a}} = \hat{z}_\mathfrak{a} + \hat{z}_{-\mathfrak{a}}
$$

If exactly one of $z_\mathfrak{a} + z_{-\mathfrak{a}}$ and $\hat{z}_\mathfrak{a} + \hat{z}_{-\mathfrak{a}}$ were zero, the above limit would evaluate as either 0 or $\infty$ instead of 1; hence this cannot be. If both are zero, Eq 43 holds without further ado. Analogously, we obtain:

$$
1 = \lim_{\substack{p \to \infty \\ p \text{ odd}}} \frac{\sum_{i \in I_\mathfrak{a}} \mathfrak{a}^p x_i + \sum_{i \in I_{-\mathfrak{a}}} (-\mathfrak{a})^p x_i}{\sum_{i \in \hat{I}_\mathfrak{a}} \mathfrak{a}^p \hat{x}_i + \sum_{i \in \hat{I}_{-\mathfrak{a}}} (-\mathfrak{a})^p \hat{x}_i} = \frac{\sum_{i \in I_\mathfrak{a}} x_i - \sum_{i \in I_{-\mathfrak{a}}} x_i}{\sum_{i \in \hat{I}_\mathfrak{a}} \hat{x}_i - \sum_{i \in \hat{I}_{-\mathfrak{a}}} \hat{x}_i} = \frac{z_\mathfrak{a} - z_{-\mathfrak{a}}}{\hat{z}_\mathfrak{a} - \hat{z}_{-\mathfrak{a}}} \quad \Rightarrow z_\mathfrak{a} - z_{-\mathfrak{a}} = \hat{z}_\mathfrak{a} - \hat{z}_{-\mathfrak{a}} \tag{44}
$$

By adding and subtracting Eqs 43 and 44, respectively, we arrive at $z_\mathfrak{a} = \hat{z}_{-\mathfrak{a}}$ and $z_\mathfrak{a} = \hat{z}_{-\mathfrak{a}}$.

**Definition 4** *Define the* canonical form *of* $\mathbf{x} \in X$ *and* $\mathbf{a} \in A$ *as*:

$$
\tilde{\mathbf{x}} := (z_{\mathfrak{a}_1}, z_{\mathfrak{a}_2}, \ldots, z_{\mathfrak{a}_s}, \underbrace{0, \ldots, 0}_{n-s \text{ zeros}}), \quad \tilde{\mathbf{a}} := (\mathfrak{a}_1, \mathfrak{a}_2, \ldots, \mathfrak{a}_s, \underbrace{0, \ldots, 0}_{n-s \text{ zeros}}). \tag{45}
$$

**Lemma 2** *Let* $\mathbf{x} \in X$ *and* $\mathbf{a} \in A$ *and* $\phi$ *be an impact function. Then* $\phi(\mathbf{x}, \mathbf{a}) = \phi(\tilde{\mathbf{x}}, \tilde{\mathbf{a}})$.

We first transform blocks of arguments to the canonical form (with some zero arguments added if necessary) step by step, and show that the value of an impact function is not affected by these transformations. The first kind of block we consider are blocks of equal non-zero parameters and corresponding non-zero abundances, i.e., $I_\mathfrak{a} =: \{i_1, \ldots, i_v\}$ for some $\mathfrak{a}$. Then, for some $\check{\mathbf{x}}, \check{\mathbf{a}}$:

$$
\phi\Big( \check{\mathbf{x}}, \mathsf{x}_{i_1} = x_{i_1} \quad , \mathsf{x}_{i_2} = x_{i_2}, \ldots, \mathsf{x}_{i_v} = x_{i_v}, \check{\mathbf{a}}, \mathsf{a}_{i_1} = a_{i_1}, \mathsf{a}_{i_2} = a_{i_2}, \ldots, \mathsf{a}_{i_v} = a_{i_v} \Big)
$$

$$
\overset{\text{D 1}}{=} \phi\Big( \check{\mathbf{x}}, \mathsf{x}_{i_1} = x_{i_1} \quad , \mathsf{x}_{i_2} = x_{i_2}, \ldots, \mathsf{x}_{i_v} = x_{i_v}, \check{\mathbf{a}}, \mathsf{a}_{i_1} = \mathfrak{a} \quad , \mathsf{a}_{i_2} = \mathfrak{a} \quad , \ldots, \mathsf{a}_{i_v} = \mathfrak{a} \quad \Big)
$$

$$
\overset{\text{I4}}{=} \phi\Big( \check{\mathbf{x}}, \mathsf{x}_{i_1} = \sum_{i \in I_\mathfrak{a}} x_i, \mathsf{x}_{i_2} = 0 \quad , \ldots, \mathsf{x}_{i_v} = 0 \quad , \check{\mathbf{a}}, \mathsf{a}_{i_1} = \mathfrak{a} \quad , \mathsf{a}_{i_2} = \mathfrak{a} \quad , \ldots, \mathsf{a}_{i_v} = \mathfrak{a} \quad \Big) \tag{46}
$$

$$
\overset{\text{D 3, I2}}{=} \phi\Big( \check{\mathbf{x}}, \mathsf{x}_{i_1} = z_\mathfrak{a} \quad , \mathsf{x}_{i_2} = 0 \quad , \ldots, \mathsf{x}_{i_v} = 0 \quad , \check{\mathbf{a}}, \mathsf{a}_{i_1} = \mathfrak{a} \quad , \mathsf{a}_{i_2} = 0 \quad , \ldots, \mathsf{a}_{i_v} = 0 \quad \Big).
$$

If a parameter $\mathsf{a}_i$ or abundance $\mathsf{x}_i$, respectively, is zero, we transform the single-index block $\{i\}$ to zero (for some $\check{\mathbf{x}}, \check{\mathbf{a}}$):

$$
\phi(\check{\mathbf{x}}, \check{\mathbf{a}}, \mathsf{x}_i = \check{x}_i, \mathsf{a}_i = 0) \overset{\text{I3}}{=} \phi(\check{\mathbf{x}}, \check{\mathbf{a}}, \mathsf{x}_i = 0, \mathsf{a}_i = 0), \tag{47}
$$

$$
\phi(\check{\mathbf{x}}, \check{\mathbf{a}}, \mathsf{x}_i = 0, \mathsf{a}_i = \check{a}_i) \overset{\text{I2}}{=} \phi(\check{\mathbf{x}}, \check{\mathbf{a}}, \mathsf{x}_i = 0, \mathsf{a}_i = 0). \tag{48}
$$

Second, after all blocks are transformed, we swap abundances and parameters in parallel to match the order in the canonical form. This does not affect the value of the impact function $\phi$ as it is commutative (I1).

**Lemma 3** *Suppose* $\mathbf{x}, \hat{\mathbf{x}} \in X$ *and* $\mathbf{a}, \hat{\mathbf{a}} \in A$ *are such that*:

$$\forall p \in \{1, \ldots\} : \quad \sum_{i=1}^{n} a_i^p x_i = \sum_{i=1}^{n} \hat{a}_i^p \hat{x}_i. \tag{49}$$

*Let* $\phi$ *be an impact function. Then* $\phi(\mathbf{x}, \mathbf{a}) = \phi(\hat{\mathbf{x}}, \hat{\mathbf{a}})$.

To prove this, we only need to note how the canonical forms $\tilde{\mathbf{x}}$ and $\tilde{\mathbf{a}}$ only depend on the parameters values $\mathfrak{a}$ corresponding to non-zero total abundance $z_\mathfrak{a}$ and these abundances. Those in turn are equal per Lemma 1. Thus:

$$\phi(\mathbf{x}, \mathbf{a}) \overset{\text{L } 2}{=} \phi(\tilde{\mathbf{x}}, \tilde{\mathbf{a}}) \overset{\text{L } 1}{=} \phi(\tilde{\hat{\mathbf{x}}}, \tilde{\hat{\mathbf{a}}}) \overset{\text{L } 2}{=} \phi(\hat{\mathbf{x}}, \hat{\mathbf{a}}). \tag{50}$$

## Non-impact-function contribution to abundance changes must be proportional

Here, we show that $\mathcal{R}_j$ as defined in Eq 10 must have the form $\mathcal{R}_j = x_j \phi_j(\mathbf{x}, \mathbf{a})$, where $\phi_j$ is an impact function. To keep the notation simple, we assume that $\mathcal{R}_j$ features no delay, noise, explicit time dependency, or similar, and thus $\mathcal{R}_j : X \times A \to \mathbb{R}$.

We require every impact of population other than $j$ to be comprised in an impact function $\psi_j$. We can write $\mathcal{R}_j$ in the form:

$$\mathcal{R}_j = \beta_j(x_j, \psi_j(\mathbf{x}, \mathbf{a})), \tag{51}$$

with $\beta_j : \mathbb{R} \times \mathbb{R} \to \mathbb{R}$. If, similar to Criterion I4, we consider the case of two populations $j$ and $k$ with identical properties and abundances $y$ and $z$, their total growth must be the same as if all individuals were assigned to one population:

$$\mathcal{R}_j(\mathbf{x}_j = y) + \mathcal{R}_k(\mathbf{x}_k = z) = \mathcal{R}_j(\mathbf{x}_j = y + z) + \mathcal{R}_k(\mathbf{x}_k = 0) = \mathcal{R}_j(\mathbf{x}_j = y + z), \tag{52}$$

where $\mathcal{R}_k(\mathbf{x}_k = 0)$ reflects that an extinct population does not grow. Using the properties of impact functions and that $j$ and $k$ are identical, we can conclude from this that:

$$\beta_j(y, w) + \beta_j(z, w) = \beta_j(y + z, w), \tag{53}$$

with $w = \psi_j(\mathbf{x}, \mathbf{a}, \mathbf{x}_j = y + z, \mathbf{x}_k = 0)$, i.e., $\beta_j$ is additive in its first argument. Under these conditions, this allows to conclude that $\beta_j$ is homogeneous (or proportional) in its first argument, i.e., $\beta_j(x, v) = x\beta_j(1, v)$ for any $v \in \mathbb{R}$. Thus the right-hand side has the form:

$$\mathcal{R}_j = x_j \hat{\beta}_j(\psi_j(\mathbf{x}, \mathbf{a})) = x_j \phi_j(\mathbf{x}, \mathbf{a}), \tag{54}$$

with $\hat{\beta}_j(v) := \beta_j(1, v)$ and some impact function $\phi_j = \hat{\beta}_j \circ \psi_j$.

## Deriving a new model for UTI strains—The legwork

We here elaborate the details of creating a new model for the case described in *Case Study*.

Inserting our ansatz (Eq 24) into our first requirement (Eq 17), we obtain:

$$0 = \mathcal{R}_j(\vec{0}, \mathbf{a}, \mathbf{x}_j = 1) = \rho_j(r_{jj})\varsigma_j(s_{jj}). \tag{55}$$

Assuming that the two factors do not "take turns" in being zero for different $j$, this means that either $\rho_j(r_{jj}) = 0$ or $\varsigma_j(s_{jj}) = 0$. Without loss of generality, we assume that the latter applies, thus assigning $\varsigma_j$ the role of quantifying the carrying capacity. Furthermore, we choose $\varsigma_j(0) = 1$. These are normalization choices, as they can be compensated by including a respective factor in $\varsigma_j$ or $\rho_j$ respectively. Using this and expanding Eq 19, we obtain:

$$0 = \mathcal{R}_j(\vec{0}, \mathbf{a}, \mathrm{x}_j = c_{jk}, \mathrm{x}_k = v) = c_{jk}\,\rho_j(r_{jj}c_{jk} + r_{jk}v)\,\varsigma_j(c_{jk} + s_{jk}v). \tag{56}$$

Assuming that $\varsigma_j$ is again responsible for the product being zero and it has only one root, namely 1, we arrive at: $c_{jk} + s_{jk} v = 1$, and thus: $s_{jk} = \frac{1 - c_{jk}}{v}$. Note that since $s_{jj} = 1$, this is consistent with our choice of $c_{jj} = 1 - v$ (see S1 Appendix).

Using the above, we can expand Eqs 16 and 18:

$$g_j \quad = \frac{\partial \mathcal{R}_j}{\partial \mathrm{x}_j}(\vec{0}, \mathbf{a}) = \rho_j(0)\varsigma_j(0) = \rho_j(0), \tag{57}$$

$$g_{jk} \quad = \frac{\partial \mathcal{R}_j}{\partial \mathrm{x}_j}(\vec{0}, \mathbf{a}, \mathrm{x}_k = v) = \rho(r_{jk}v)\varsigma(v s_{jk}) \quad \Rightarrow \quad \rho(r_{jk}v) = \frac{g_{jk}}{\varsigma(s_{jk}v)} = \frac{g_{jk}}{\varsigma(1 - c_{jk})}. \tag{58}$$

We choose the arguably simplest function to fulfill the criteria for $\rho$, namely $\rho_j(z) := g_j + z$. This has the consequence:

$$\rho_{jk} = \frac{1}{v}\left(\frac{g_{jk}}{\varsigma(1 - c_{jk})} - g_j\right). \tag{59}$$

A group of functions fulfilling the criteria for $\varsigma$ is: $\varsigma_j(z) := 1 - \lceil z\rceil^q$ with $q > 0$ and $\lceil z \rceil := \max(0, z)$. Here, the free parameter $q$ controls how early and smoothly the saturation effect of a occupied niche kicks in. Note that this choice results in terms similar to what was named *hyperlogistic* [72].

Finally, like the original study [16], we constrain the growth and capacity term to be nonnegative to avoid the occasional implausible result. For example, we do not allow negative growth because we equate the abundance of a population with its footprint, which cannot be undone, and we lack the data to capture cell death. Putting everything together, we arrive at the model:

$$\dot{x}_j = x_j\left[g_j + \sum_{i=1}^{n}\frac{1}{v}\left(\frac{g_{ji}}{1 - \lceil 1 - c_{ji}\rceil^q} - g_j\right)x_i\right] \cdot \left[1 - \left\lceil\sum_{i=1}^{n}\frac{1 - c_{ji}}{v}x_i\right\rceil^q\right]. \tag{60}$$

A problem with this model is that for $0 < x_k < 1$, we have: $\lim_{c_{jk}\to 0}\dot{x}_j = \lim_{c_{jk}\to 0}\mathcal{R}_j(\mathbf{x}, \mathbf{a}) = \infty$. Now, $c_{jk} = 0$ means that there is no growth of strain $j$ in the medium conditioned by strain $k$ and thus we already have a problem with experimentally determining $g_{jk}$. Thus, one might argue that the actual point of the singularity requires a dedicated case distinction anyway. However, $\lim_{c_{jk}\to 0}\dot{x}_j = \infty$. also means that $\dot{x}_j$ becomes arbitrarily large for small $c_{jk}$. A way to address this problem is to consider the case $q \to \infty$, or more specifically:

$$\lceil\varsigma_j(z)\rceil = \begin{cases} 1 & \text{if } z < 1 \\ 0 & \text{if } z \geq 1 \end{cases}. \tag{61}$$

In this case, the term $\varsigma(1 - c_{jk})$ in Eq 58 can be assumed to be 1 (otherwise, we would have the aforementioned problem of not being able to experimentally determine $g_{jk}$). This eliminates the singularity, but also renders the model not continuously differentiable.

In our simulations, we therefore make a trade-off between complying with Eq 18 and the numerical benefits of a continuously differentiable model by setting $q = 10$ and approximating $\varsigma(1 - c_{jk}) \approx \lim_{p \to \infty} \varsigma(1 - c_{jk}) = 1$ in Eq 58, thus arriving at:

$$\dot{x}_j = x_j \left[ g_j + \sum_{i=1}^{n} \frac{g_{ji} - g_j}{\nu} x_i \right] \cdot \left[ 1 - \left[ \sum_{i=1}^{n} \frac{1 - c_{ji}}{\nu} x_i \right]^{10} \right]. \tag{62}$$

## Supporting information

**S1 Appendix. Further analysis of UTI models.**
(PDF)

**S1 Code. Code for the simulations for Fig 1 and S1 Appendix in a tarball.**
(TAR)

## Acknowledgments

We are grateful to H. Arndt, J. Aufdermauer, M. Cosentino-Lagomarsino, A. Espinosa-Cantú, U. Feudel, J. Freund, S. Khaiwal, Y. Mulla, G. Petrungaro, S. Vet, M. de Vos, J. Werner, and M. Zagorski for inspiring discussions or constructive comments on previous versions of the manuscript.

## Author Contributions

**Conceptualization:** Gerrit Ansmann, Tobias Bollenbach.

**Data curation:** Gerrit Ansmann.

**Formal analysis:** Gerrit Ansmann.

**Funding acquisition:** Tobias Bollenbach.

**Investigation:** Gerrit Ansmann.

**Methodology:** Gerrit Ansmann.

**Software:** Gerrit Ansmann.

**Supervision:** Tobias Bollenbach.

**Visualization:** Gerrit Ansmann.

**Writing – original draft:** Gerrit Ansmann.

**Writing – review & editing:** Gerrit Ansmann, Tobias Bollenbach.

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
