## [Decision Letter · Decision Letter 0]

25 Sep 2020

Dear Dr. Ansmann,

Thank you very much for submitting your manuscript "Building clone-consistent ecosystem models" for consideration at PLOS Computational Biology.

As with all papers reviewed by the journal, your manuscript was reviewed by members of the editorial board and by several independent reviewers. In light of the reviews (below this email), we would like to invite the resubmission of a significantly-revised version that takes into account the reviewers' comments.

We cannot make any decision about publication until we have seen the revised manuscript and your response to the reviewers' comments. Your revised manuscript is also likely to be sent to reviewers for further evaluation.

Sincerely,

Kiran Raosaheb Patil, Ph.D.

Associate Editor

PLOS Computational Biology

Jason Papin

Editor-in-Chief

PLOS Computational Biology

Reviewer's Responses to Questions

**Comments to the Authors:**

Reviewer #1: Reproducibility report has been uploaded as an attachment.

Reviewer #2: A basic requirement of any mathematical model is that it should provide consistent results if applied to the same situation. While this would have been of intuitive to the founders of theroretical ecology, it seems to be increasingly forgotten in more recent models that can accomodate a flexible number of species. In those models one could arbitrarily split a species into two model species and would expect the same result as if they were represented by one model species. But not all current models meet this essential requirement. The authors present a novel framework by which this requirement can be met in practice.

Many readers will feel that they know and use these concepts already, but a large enough fraction of modellers seem to be unaware of it that this publication is valuable.

The manuscript leaves very little to criticise in terms of content or presentation and would be publishable in the present form.

However, let me mention a subtle point: The author's want to make sure they avoid the impression of being naive with regards to the intentions of model designers.

Suppose I want to model a number of stably coexisting populations that compete for a common resource. I might write

dR/dt = Rin - sum (a_i R X_i )

dX_i/dt = a_i X_i R - X_i^2

Of course this model violates the clone consistency conditions. The authors would argue that I have made a mistake and should have used loss terms of the form X_i (sum X_j) instead of X_i^2. This is in principle a valid point and many readers will profit from seeing it. However, for the sake of argument I want to defend the X_i^2 terms. I stated above that I want to model stably coexisting populations. We know that asymptotically stable coexistence is only possible if the different species occupy distinct niches (modeled in this case by the square terms). In this case the clone consistenty argument formally does not apply. I cannot argue that I can arbitralily split a population in two and represent it by two model variables, because the resulting populations would occupy the same niche and are hence outside the scope of my model.

As another example: If this were taken to far one could criticise Bob May's famous 1972 random matrix model for not being clone consistent. The model describes species interactions by a random matrix where elements are drawn independently. But if I would split a species in two arbitrarily the corresponding rows and columns would be correlated.

My point is that if I intend to model populations in different niches, this intention alone already removes the need for clone consistency in the model. (Conversely of of course one should be aware that violating clone consistency as doing so inserts arbitrary niches and thus constitutes an implicit assumption as the authors point out.).

The takeaway from is that one cannot decide whether a model is good or bad by looking at the equations alone, because whether clone consistency is even a criterion depends on the detailed premises of the model. The authors are right when they point out it would be bad to violate clone consistency accidentally without being aware of the additional assumptions that this implies. But, on the other hand, I would be equally bad to enforce clone consistency when it is at odds with the explicitely states assumptions. Doing so would prevent us from exploring some very valuable mathematical models.

The wording in some places suggests that the authors are aware of this point, but I fear that it would be lost on the casual or less informed reader will probably miss this.

It would be good to mention it prominenently (e.g. when the authors present their introductory example) and perhaps hint in the abstract and/or author summary that sometimes clone inconsistecy is intentional and very much desired.

Summary: This is a good paper. It should be published, but please revisit the phrasing. There are good reasons to violate clone consistency in models and models which do this must still be permissble in the future. If this paper turns out to give future authors unwarranted trouble with reviewers of the future then more harm than good might be done.

Best regards

Thilo Gross

Reviewer #3: In their paper, the authors address the problem that in microbial community models, where the interacting species cannot be guaranteed to be sufficiently different from each other, models need to be tailored to account for this. The authors develop a framework of constructive properties for dynamical systems for the population dynamics, i.e. the right hand side of the corresponding differential equations, which guarantee that aggregation and splitting of state variables preserve the dynamics of the model. The authors call this preservation property clone consistency. The authors provide guidelines on how to use their framework to build consistent models and on how to detect if a given model is consistent. They also explain how to resolve inconsistencies in models within the limits of biological plausibility. To develop the framework the authors show that to be consistent the RHS must be the product of the abundance of the focal species and a term that contains all effects that other species in the community have on the focal species. The authors call this term an impact function. They show that an impact function must be composed of basic building blocks: linear combinations of interaction parameters and population abundances. Furthermore, the authors show that the impact functions are contained in a functional algebra, which implies that combining impact functions in prescribed ways will lead again to impact functions. The authors apply their framework to build a clone consistent version of a previously published model of microbial interactions in urinary tract infections.

I consider the constructive framework developed by the authors and its translation into instructions on how to build consistent models, a valuable contribution to the field of ecological population models. However, I have some major concerns regarding the presentation of the work, its lack of context with respect to previous results in this area and the clarity and completeness of the interpretation of the results in the context of nonlinear dynamical systems. These major issues would need to be addressed for the article to proceed. They are detailed in the following together with several minor points. Numbers in parenthesis refer to the line number in the manuscript.

Major issues

Issue 1:

While the constructive framework appears to be plausible, the soundness of its justification suffers from unclear language, making it difficult to follow. I advise the authors to get feedback from colleagues with little familiarity with the work or even to work with a writing coach to improve the flow and readability of the text. The language should also be revised in terms of scientific writing standards. For example, I do not believe ‘inevitable’ to be the proper vocabulary to describe properties of mathematical objects. Furthermore, the authors should decide on a target audience and reflect this in their presentation. The presentation is uneven in terms of the required background knowledge in order to comprehend the statements.

Examples are provided under ‘Examples’ below.

Issue 2:

While the authors provide accessible guidelines on how to construct a consistent model and how to check existent models for consistency, the manuscript fails to state and clarify the mathematical properties that allow models of the proposed form to be consistent. In the authors summary the authors claim to have ‘investigated the mathematical properties of clone-consistent models’, but I do not see where in the manuscript these mathematical properties are described. The authors do provide criteria that impact functions should fulfil, but never explain why they do so or actually show that they do. In addition, how the properties of the impact functions translate to the properties of the whole model and especially the solutions of the associated differential equations is left unclear. Furthermore, the authors state not to be able to make statements about the cause of inconsistency (line 392). I am wondering how the cause of inconsistency is connected to the fact that a nonlinear differential equations is unlikely to show additivity in the systems state variables (in the sense that f(x)+f(y) is not equal to f(x+y)) unless it is built in a way that is has this property? Just as the actual cause of the inconsistency in the motivating example on lions is not that logarithms are not allowed in impact functions but that these system equations are not additive in x_2 and x_3. In summary, this comes down to a lack of the discussion of the framework in the context of nonlinear dynamical systems, which is also closely connected to the next issue.

Issue 3:

The manuscript fails to address how the findings relate to previous research in the area of nonlinear differential equation model reduction based on the aggregation of several state variables into a single one. If the goal is to preserve the system dynamics exactly this is for example termed ‘perfect aggregation’ in the field of ecological modelling, or ‘exact lumping’ in the field of biochemical reaction network modelling (just to mention two out of the many fields). The aggregation of variables via sums, as also considered by the authors, is a special case of what is called aggregation by a linear map or linear lumping in the context of the aforementioned fields, respectively. The following references provide an overview on this topic but are by no means exhaustive:

Tóth, J., Li, G., Rabitz, H., & Tomlin, A. S. (1997). The effect of lumping and expanding on kinetic differential equations. SIAM Journal on Applied Mathematics, 57(6), 1531–1556. https://doi.org/10.1137/s0036139995293294

and references therein, especially

Iwasa, Y., Andreasen, V., & Levin, S. (1987). Aggregation in model ecosystems. I. Perfect aggregation. Ecological Modelling, 37(3–4), 287–302. https://doi.org/10.1016/0304-3800(87)90030-5

and references therein.

I am aware that the framework proposed by the authors goes beyond the aggregation of identical species to also include consistency regarding the aggregation of identical modes of interaction only, but I still believe that the manuscript would profit from a discussion of the connection to the above-mentioned work.

Examples

Lack of reader guidance:

In general, the manuscript in several places lacks paragraphs to guide the reader. With that, I refer to paragraphs especially in the beginning of a section that prepare the reader for what results to expect in this section and which plot out the strategy on how the authors will get there. A general ‘the paper is structured as follows …’ paragraph mentioning what to find in which section would also be helpful. For example, the section ‘What models are clone consistent’ right away starts out with impact functions without any information on how these functions relate to the models that the authors set out to find. Other examples are ‘The Functional Algebra of Impact Functions’ (490) and ’Deriving a New Model for UTI strains - the Legwork’ (598).

Problems in manuscript structure:

Figure 3 cannot be understood since needed information is only given later in the manuscript. Upper part of the figure: The reader cannot understand why question 5 is answered the way it is, because the checked model is only introduced and discussed later in the manuscript. This is especially a problem for question 5, since the answers to this question cannot be interpreted without the information given later in the manuscript. The answers to question 5 are confusing as well since it is not clear whether the answer is yes (no) or depending on the application. Finally, when the question of ecological plausibility is dealt with in the manuscript, the connection to question 5 (or Figure 3) is not indicated (line 294 ff). The authors also might want to reconsider the relevance of question 4, since later in the manuscript (line 340) it is stated that rewriting an equation in this way is always possible. Lower part of Figure 3: The meaning of the footnote within the figure is unclear. In question 2 omega which was previously defined as a basic impact function is used as symbol for a function of a basic impact function.

Inconsistency between section title and section content:

The section ‘Implications contains’ contemplation on various aspects of which most content wise fit in neither of the subsections titled ‘Checking Clone Consistency to Reveal Implicit Assumptions’ and ‘General Implications for Model Design’.

Ambiguities making it hard for the reader to grasp the essential concept:

The models described by equations (2) and (3) cannot be the ‘same model’ (line 68 and caption of Figure 1) since one of them has two state variables while the other on has three.

Incomprehensive paragraphs:

The paragraph starting from line 217 on building models with aggregated phenomenological observables:

• 223: ‘each of the m experimentally determined interaction parameter’. Not clear what is meant with interaction parameters here. Each single measurement? Interaction parameter seems to have a different meaning here than in line 42.

• What is the reasoning behind the general Ansatz in equation (13)? The section on how to determine parameters and functions (224ff) is unclear.

• 231: What are building blocks in this context?

• 234: ‘Finally, for some applications, a sum or more complex way to combine the basic impact functions may be appropriate (as opposed to the product used in Eq. 13). In New Model, we provide an example for this approach.’

What approach? The product version or the sum?

Non-causal relations between sentences that imply such relations:

Line 47: ‘These new experimental scenarios often call for new ecological models that can incorporate the respective data. One reason for this is that there is no single answer as to how multi-parameter o higher-order interactions should be measured [3, 6, 16, 20, 23].’

Overall meaningless sentences like: ‘Moreover, clone-inconsistent models are diverse for the same reason that non-linear functions are.’ (394)

Minor issues

Main text

1. It is not clear to me why the authors come up with a new label (clone consistency) for something that has been described before. The authors state mere similarity to earlier descriptions of the problem (line 59). If there is indeed a difference between earlier descriptions of the problem and ‘clone consistency’, I would like to request a discussion of those differences by the authors to justify new wording.

2. It is not clear if the proposed framework can also be used for models that model the mechanism of interaction, e.g. for consumer resource models. If so, how does the proposed framework integrate with commonly used impact functions? The authors should unambiguously state in the introduction the latest for which types of models this framework can be used, so readers can decide early on whether the paper is relevant for their research.

3. 115, 123, 203,… the Volterra model [1]): not clear to which model/equation in this reference the authors refer to. What is the difference between what the authors call the Volterra model and what is commonly referred to as the generalized Lotka-Volterra model?

4. In the authors summary the authors state that something along the lines of that one way to test if a models make sense is to check if the dynamics stay the same if you split a model species. I do not agree that this is either a necessary or a sufficient condition to decide on the validity of a model.

5. The authors should clarify how their results for the system translate to properties of fixed points

Methods

6. As indicated to the comment on reader guidance, the methods part would also benefit from more supportive text to help the reader understand what to expect from the section. The subsection about notation indicates a comprehensive symbolic language that is used rigorously. This is not the case e.g.:

1. the use of mathfrak symbol sequences in Definition 1.

2. the quantifiers are used inconsistently, i.e. after an expression in (30), before an expression in (38)

3. The appearance of the triple bar symbol in (40)

7. The overall presentation indicates a formal approach by applying a Definition-Theorem-Proof style. This is not consistently applied, e.g.:

1. the domains and codomains of the impact functions or the functions \\mathcal{R} are never given

2. the informal introduction of the terminology "point-separate" which appears to be the same as "separate" in functional analysis textbooks

8. The section ‘Non-Impact-Function Contribution to Abundance Changes Must be Proportional’ seems to reproduce the textbook argument that additivity plus continuity implies homogeneity, see e.g. Linearity on Wikipedia. It leaves out the decisive steps from that argument. Because of both, the involved way of going to the linearity in the first argument and the obscure presentation, the value of the paragraph for the reader is unclear. Without already knowing what the section is about, the section title is hard to interpret.

9. Paragraph from 505- 509. The authors need to clarify how this is connected to what the reader has learned about impact functions up to this point. What is meant by ‘capture the application’?

S2 Comparing the Old and New UTI Model with Experimental Data

10. The statement “a strain cannot grow on the portion of the medium that is its own supernatant” (line 677) is not generally true. Without the respective experiments, this statement cannot be made (a fact that the authors seem to acknowledge in line 255). If a species has grown to saturation in a medium this does not necessarily mean that growth saturated due to complete exhaustion of an essential nutrient. Growding effects for example also play a role. If the authors need to use this as an assumption in their model they need to formulate is as such.

11. 680: ‘Second, in the medium conditioned by itself, a strain's growth rate should at best slightly lower than in an unconditioned medium …’

This is not a generally valid statement. It depends very much on whether nutrients are in excess in your medium or not. As long as the nutrients in the mix of conditioned and new medium are not limiting the maximum relative growth rate will also be reached under these conditions. The authors should clarify how the g_{ii} were chosen within the mentioned interval (line 683) for the simulations.

12. 683: ‘Without these adjustments, we would obtain implausible results, e.g., in case of cii > 1 - v, the respective strain could never stop growing since it effectively increases the size of its own niche.’

The authors should clarify this sentence. It is not clear to me what the authors mean here with adjustments.

13. The findings regarding better or worse agreement with the data in comparison to the old model are not comprehensible. Neither the measure based on which these statements are made nor the features that were used to assess agreement are stated. Claims about the stability of fixed points are made without a suitable analysis. Causes for observed differences and agreements in model outcomes are put forward without justification through simulations. There is no way to compare the population dynamics of the two models since only simulation endpoints are shown.

14. Caption of Figure 6: ‘All simulations converged, except for the model from Ref. 16 and Community 7, which exhibited strong chaotic fluctuations.’

The authors should clarify what they mean with converged and with chaotic fluctuations/ oscillations.

S3 files

15. The authors should provide a README file together with the data and code files. This file should state the content of each file (e.g. which is the file that contains the data that were declared to be available in these files and which file was used to create which figure).

Misspellings and miswording

The term 'point-separates' appears to be a literal translation of the German wording, please check whether this is consistent with the contemporary term in English.

‘Shape’ of equations (functions, models, … ) (144, 147, 151…). The shape of an equation refers to the graphical representation of an equation. In the context that shape is used here, form would be the correct word (c.f. standard form of an equation, factored form of an equation, …)

An abstract object like a mathematical model cannot have a biological shortcoming (390) since it does not have a biology.

‘Capture the application’ (line 504) seems to be a literal translation of the German wording, unclear what is meant here.

Line 73: predators -> predator

Line 260: this data -> these data (data are plural)

Line 387: targeting -> targeting

Line 417: general modelling -> generalized modelling (according to the provided reference)

Equation 35: limits of the sum missing in last equation?

512: There is no j in the equation that needs to be defined, but I think the definition for p is missing

520: unitial -> unital

524: unitiality -> unitality

650: represent -> represents

681: best slightly -> best be slightly

**Have all data underlying the figures and results presented in the manuscript been provided?**

Reviewer #1: None

Reviewer #2: Yes

Reviewer #3: Yes

PLOS authors have the option to publish the peer review history of their article (what does this mean?). If published, this will include your full peer review and any attached files.

Reviewer #1: No

Reviewer #2: **Yes: **Thilo Gross

Reviewer #3: No
---

## [Decision Letter · Decision Letter 1]

30 Nov 2020

Dear Dr. Ansmann,

Thank you very much for submitting your manuscript "Building clone-consistent ecosystem models" for consideration at PLOS Computational Biology. As with all papers reviewed by the journal, your manuscript was reviewed by members of the editorial board and by several independent reviewers. The reviewers appreciated the attention to an important topic. Based on the reviews, we are likely to accept this manuscript for publication, providing that you modify the manuscript according to the review recommendations.

Sincerely,

Kiran Raosaheb Patil, Ph.D.

Associate Editor

PLOS Computational Biology

Jason Papin

Editor-in-Chief

PLOS Computational Biology

[LINK]

Reviewer's Responses to Questions

**Comments to the Authors:**

Reviewer #2: The authors have addressed my previous concerns. This is a valuable paper. I recommend publiction.

Reviewer #3: In the revised version of the manuscript, most of my issues have been addressed. However, in the following I try to clarify one of my original issues that I think has not been fully addressed. Furthermore, I list some points where I disagree and some that are still not fully clear to me.

'We do not expect that clone consistency or the use of impact functions allows for drawing general conclusions about the respective models, including about their dynamics.'

The conclusion that in a clone consistent model the dynamics must stay the same if populations are merged (or split) is a conclusion about the model dynamics. My point comes down to the problem that the manuscript discusses clone consistency in two different contexts:

1. Clone consistency as a property of impact functions, which is defined verbally (I4 main text) and formally in the paper (I4in Methods).

2. Clone consistent population models, i.e. clone consistency as a property of a population model (eqn. (10)). The criterion for clone consistency in population models is discussed verbally for example in the abstract and illustrated in the motivating example. Clone consistency of models is not defined formally.

The manuscript draws connections between impact functions and population models, e.g. eqn. (11). How the clone consistency of impact functions determines the clone consistency of population models is not explained, because clone consistent population models are not defined. The authors should clarify this connection.

'We now use general model instead of model to make clear that this refers to Eq. 1 (in which the number of populations is a parameter) and not a specific realisation of this general model (as described by Eqs. 2 and 3). We refrained from removing this formulation altogether since the fact that the simulations are based on the same general model is crucial here and this wording clearly communicates this.'

I am not convinced that the addition of the word ‘general’ makes this clearer. What’s the purpose of showing specific realisations if those do not relevant for the example? In my opinion, this could be easily made clearer by using only Eq. 3 (instead of Eqs. 2+3). Then, to distinguish between the two scenarios only the vector of initial conditions is needed in addition to equation 2.

243: First we identify the number of basic impact functions m as the number of experimental interaction observables, i.e., the number of measurements per (ordered) pair of populations.

This is sentence still has the potential to confuse a first time reader. The first part of the sentence is clear but in the second part ‘number of measurements’ sounds like it has something to do with replicates. I think it would be clearer to just give an example for an observables in addition to the first part of the sentence.

Footnote in Figure 3: “This part only applies to models of population growth as in Eq. 10”.

I still do not get what this footnote tries to tell the reader. The figure panel is about building a model. So what is the alternative to a "model of population growth" here? When would the parts indicated by the footnote not apply?

In line 334 the authors state that TWO experimental interaction parameters are available, but equations 16 -19 are FOUR criteria that are referred to in 304 as ‘one criterion per experimental interaction parameter’. Are there two or four interaction arameters? This seems still to be related to the ambiguous use of the term ‘interaction parameter’ in the manuscript. On the one hand used for modes of interaction (interaction observables?) and on the other hand referring to the actual parameter in an equation. The authors should also explicitly state in words for each criterion (Eq. 17-19) to which ‘interaction parameter’ it corresponds.

315 As the model clearly satisfies Criteria I1-I3, it therefore must violate I4…

Where is the causality between these two parts of the sentence?

358: Moreover, the fixed points of both models are usually the same.

What is meant by usually here?

543: Otherwise \\Xi would generate more than impact functions.

Not clear what ‘more than impact functions’ could be.

242: Here, we refrain from transforming the parameters in Eq. 14

Should this be Eq. 13? Otherwise, it would make sense to make this statement after Eq. 14.

**Have all data underlying the figures and results presented in the manuscript been provided?**

Reviewer #2: Yes

Reviewer #3: Yes

PLOS authors have the option to publish the peer review history of their article (what does this mean?). If published, this will include your full peer review and any attached files.

Reviewer #2: No

Reviewer #3: No
---

## [Editor Report · Decision Letter 2]

15 Dec 2020

Dear Dr. Ansmann,

We are pleased to inform you that your manuscript 'Building clone-consistent ecosystem models' has been provisionally accepted for publication in PLOS Computational Biology.

Best regards,

Kiran Raosaheb Patil, Ph.D.

Associate Editor

PLOS Computational Biology

Jason Papin

Editor-in-Chief

PLOS Computational Biology

---

## [Editor Report · Acceptance letter]

29 Jan 2021

PCOMPBIOL-D-20-01380R2 

Building clone-consistent ecosystem models

Dear Dr Ansmann,

I am pleased to inform you that your manuscript has been formally accepted for publication in PLOS Computational Biology. Your manuscript is now with our production department and you will be notified of the publication date in due course.

With kind regards,

Alice Ellingham
